# Relationship between Insufficient Sleep and Bad Breath in Korean Adolescent Population

**DOI:** 10.3390/ijerph17197230

**Published:** 2020-10-02

**Authors:** Kyung-Yi Do

**Affiliations:** Department of Dental Hygiene, Hanseo University, 46 Hanseo 1-ro, Haemi-Myun Seosan-si, Chungcheognam-do 31962, Korea; dkl8684@naver.com

**Keywords:** insufficient sleep, halitosis, adolescents, Korean youth risk behavior web-based survey (KYRBS)

## Abstract

This cross-sectional study sought to evaluate the association between insufficient sleep and bad breath among Korean adolescent population. It was based on the 13th Korea Youth Risk Behavior Web-Based Survey (2017). From 64,991 participants (aged 13–18 years), the final participation rate in the survey was 95.8% (62,276 participants; 31,624 boys and 30,652 girls). A complex sample logistic regression was performed to identify the relationship between insufficient sleep and halitosis, after adjusting for all covariates. In Model II for estimating the adjusted odds ratio (AOR) for general characteristics, students who answered “not at all sufficient”, indicating insufficient sleep, were at higher risk of bad breath than those who answered “completely sufficient” (AOR = 2.09, 95% confidence interval, CI = 1.91–2.30). In Model III, for estimating the AOR adjusted for all covariates, students who answered “not at all sufficient”, indicating insufficient sleep, were at higher risk of bad breath than those who answered “completely sufficient” (AOR = 1.47, 95% CI = 1.33–1.83). It is necessary for families and schools to have health education lessons that recognize insufficient sleep among adolescents may be a cause of bad breath and therefore optimal sleeping habits and oral health behaviors should be promoted.

## 1. Introduction

Sufficient sleep facilitates optimal human metabolism and plays an important role in maintaining the normal functioning of the body and brain. Sleep restores damaged cells and muscles, relieves tiredness and fatigue, and facilitates well-being and a healthy life [1,2,3,4]. In contrast, if sleep duration and quality is insufficient, this may cause and exacerbate fatigue, stress in daily life, and decreased concentration, and may contribute to the development of chronic diseases such as diabetes, hypertension, and heart disease [3]. Lack of sleep also increases susceptibility to infections owing to changes in human hormones, reduced immunity, and inflammatory biomarker expression [1,2,3,5].

Lack of sleep is also closely related to oral health [6]. Sleep deprivation results in reduced saliva flow and IgA secretion rate; additionally, oral health is associated with insufficient sleep, increased salivary interleukin-6 production, and high levels of *Streptococcus mutans* colony counts, which could be a factor contributing to dental caries [7,8,9,10]. A decreased rate of salivary flow in the mouth during sleep may decrease the antibacterial and cleaning effects of saliva and render teeth vulnerable to dental caries. An unpleasant odor may occur because of food residue between the teeth and putrefaction of the retained exfoliated oral epithelial cells, and this may increase the plaque accumulation on the teeth and tongue, resulting in more severe bad breath [10,11]. In a study of children and adolescents aged 12–16 years by Guedes et al. (2019), subjects with dental caries had a 3.8-fold higher risk of bad breath than those without, and decreased salivary flow increased the risk of bad breath 4.2-fold times [12].

The average sleep time of adolescents reported by the Korea Centers for Disease Control and Prevention (KCDCP) was much less than the sleep time recommended by The National Sleep Foundation (8.5–9.25 h for persons aged 12–17 years, not including naps). It has been reported that sleep time decreases from elementary school to higher grades. In particular, Korean high school students report that their average sleep time is only 6.1 h, and this is of concern and requires attention and intervention [13,14,15].

The main causes of bad breath are oral-related pathological factors such as oral cavity, periodontitis, poor oral hygiene, tongue coating, lack of flow of saliva during sleep, salivary gland dysfunction, dry mouth, poor dietary habit, and smoking. Other factors include leukemia, diabetic ketoacidosis, respiratory system, stomatitis, and medications [11,16,17]. The prevalence of bad breath may vary according to the country, ethnicity, sex, age, dietary habits, socio-cultural differences, and research methods. The worldwide rate of halitosis is between 10% and 50%, and the prevalence is high in adults as well as in children and adolescents [16,18]. The KYRBS (2017) reported that over 20% of adolescents answered that they had an unpleasant smell in their mouth, suggesting that the prevalence of bad breath in Korean adolescents is extremely high [19]. Bad breath can have a negative psychological and social effect on adolescents and can be a sign of other diseases [11,12]. Therefore, it is crucial to identify the cause of bad breath in adolescents and make appropriate interventions.

Existing studies related to bad breath (halitosis) mainly focus on adults, and various causes have been identified [12,20]; however, studies on the relationship between insufficient sleep and bad breath are rare. Therefore, this study investigated the relationship between insufficient sleep and bad breath in adolescents based on national data representing a sample of Korean adolescents, with a view to recognizing bad breath as a disease and encouraging the promotion of oral health in adolescents through early diagnosis and appropriate treatment.

## 2. Materials and Methods

### 2.1. Participants

This study was based on the 13th Korea Youth Risk Behavior Web-based Survey (KYRBS, 2017), which is conducted annually by the Ministry of Health and Welfare, Ministry of Education, and Korea Centers for Disease Control and Prevention—on South Korean adolescents’ health behaviors. To establish the sample frame, data collected from middle and high schools across the country as of April 2016 were used. The sampling procedure used can be divided into population stratification, sample allocation, and sampling. In population stratification, to decrease the sampling error, 39 regions and three school types (middle school, general high school, and specialized vocational high school) were used as stratification variables to divide the population into 117 subpopulations. In sample allocation, the sample size was determined as 400 middle schools and 400 high schools. For 17 cities and provinces, five middle schools and five high schools were allocated first. Stratified cluster sampling was used with the school as the primary extraction unit, and the class was used as the secondary extraction unit. Among classes selected in the secondary sampling, one class was selected randomly and all students in the class were selected. Those who had been absent for prolonged periods, those with special needs, and those impaired literacy were excluded. More details about the nationwide survey sampling and process are available elsewhere (KCDCP, 2017).

The survey was performed with 64,991 students from 400 middle and 400 high schools across South Korea. The final participation rate was 95.8%, with data from 62,276 students from 799 schools collected (31,624 males and 30,652 females, age range 13–18 years). The KYRBS, a government-approved survey (approval number 117058), was reviewed by the institutional review board of the Korea Centers for Disease Control and Prevention and conducted upon obtaining consent from the participants.

### 2.2. Measurements

The KYRBS was conducted anonymously on middle and high school students through self-reported online questionnaires. The survey included 123 questions in 15 areas. Some of these areas included the following: health behaviors, oral health, mental health, internet addiction, smartphone use, smoking, drinking, obesity, physical activity, and eating habits. This study extracted and used questions related to bad breath factors (one factor), health behavior-related factors (six factors), and general characteristics (seven factors), from the 13th KYRBS questionnaire (2017), and the variables were re-categorized according to the purpose of this study (KCDCP, 2017).

#### 2.2.1. General Demographic Characteristics

General characteristics, including sex, grade, academic achievement, perceived family economic status, living status, and father’s and mother’s education, were collected. Grade was classified into middle school or high school, and these two categories were further classified into 1st/2nd/3rd grades. Academic achievement and economic status were classified into “high” “middle”, and “low” (originally “high”, “upper-middle”, “middle”, “lower-middle”, and “low”). Living status was categorized as “living with family”, living with relatives”, “boarding”, or “living in dorm (including care facility)”. Parents’ education level was classified into “middle school and below”, “high school”, “university (including college)”, or “unknown”.

#### 2.2.2. Health-Related Behaviors

Health-related behaviors, including alcohol consumption and smoking history, stress level, frequency of teeth-brushing, teeth-brushing after lunch, and insufficient sleep (fatigue recovery after sleep), were obtained. Alcohol consumption was determined by the “yes” or “no” in response to the question, “Have you ever had more than one glass of alcohol?”. Smoking history was determined by the “yes” or “no” response to the question, “Have you ever inhaled from a cigarette even once or twice?”. Stress level was determined by responses comprising “very much, a lot” “a little” “not much”, and “not at all” to the question, “In general, how much stress do you feel?”. Teeth-brushing frequency of the previous day was divided into “none”, “once a day”, “twice a day”, “three times a day”, or “four or more times a day.” Teeth-brushing after lunch at school was divided into “always”, “almost always”, “sometimes”, and “never”.

#### 2.2.3. Insufficient Sleep

Insufficient sleep as an independent variable was categorized as “completely sufficient”, “sufficient”, “moderate”, “not sufficient”, and “not at all sufficient”, in response to the question, “Was the amount of sleep you had in the past 7 days sufficient for relieving fatigue?”.

#### 2.2.4. Bad Breath

Finally, bad breath as a dependent variable was classified according to the response to the question, “Have you experienced bad breath in the last 12 months?” as “yes” or “no.”

### 2.3. Statistical Analysis

All analyses were conducted using a complex sample analysis module considering the stratification variable, cluster variable (PSU), and sample weight. First, we conducted the Rao–Scott χ^2^-test to present an N (weight %) and a standard error to examine whether there was a significant difference in the distribution of bad breath according to the participants’ general characteristics and health-related behaviors. Second, we performed a complex sample logistic regression analysis and presented the odds ratios (ORs) and 95% confidence intervals (CIs) to identify the relationship between insufficient sleep and bad breath. Model I investigated the relationship between insufficient sleep and bad breath (unadjusted). Model II investigated the relationship between insufficient sleep and bad breath after adjusting for general characteristics. Model III investigated the relationship between insufficient sleep and bad breath after adjusting for all covariates (sex, grade, academic achievement, perceived family economic status, living status, father’s education, mother’s education, alcohol, smoking, stress level, frequency teeth-brushing, and teeth-brushing after lunch). Statistical analyses were conducted using PASW statistics 18.0 ver. (IBM Co., Armonk, NY, USA). Significance was set at <0.05.

## 3. Results

### 3.1. Bad Breath According to General Characteristics

The prevalence of bad breath in boys and girls was similar (boys 22.5% vs. girls 22.0%). Middle school second students had the highest prevalence of bad breath (23.3%, *p* = 0.008). The lower the academic achievement, the higher the prevalence of bad breath (25.5%, *p* < 0.001), and the lower the economic status, the higher the prevalence of bad breath (32.2%, *p* < 0.001). In terms of living status, the highest prevalence of bad breath was among students who reported living in boarding and dorms (including care facility) (24.8% *p* = 0.006). In terms of father’s education, 31.3% had achieved middle school or lower, and in terms of mother’s education, 31.2% had achieved middle school or lower; the lower the education, the higher the prevalence bad breath (*p* < 0.001, Table 1).

### 3.2. Bad Breath According to Health Behaviors

The rate of bad breath was slightly higher in students with a history of alcohol consumption than in students with no drinking experience (22.0% vs. 21.2%). Students with smoking experience had a higher rate of bad breath than students without smoking experience (24.5% vs. 21.9%, *p* < 0.001). The stress level was about three times higher among students who answered “very much” than those who answered ‘‘not at all” (30.1% vs. 11.6%, *p* < 0.001). Frequency of teeth-brushing per day in students who answered “none” was about 2.5 times higher than among those who answered “3 times a day” (18.3% vs. 45.4%, *p* < 0.001). There was a higher prevalence of bad breath among students who answered “never” than among those who answered “always” to the question about teeth-brushing after lunch at school (25.3% vs. 17.2%, *p* < 0.001). Finally, in terms of the relationship between insufficient sleep and h bad breath, students who reported “not at all sufficient” sleep had a higher bad breath rate than those who answered “completely sufficient” sleep (28.3% vs. 16.2%, *p* < 0.001, Table 2).

### 3.3. Logistic Regression Analysis of the Association between Insufficient Sleep and Bad Breath

A complex sample logistic regression was performed to examine the association between insufficient sleep and bad breath. In Model I, which estimated the crude OR before adjustment for covariates, higher crude ORs were found in students who answered “not at all sufficient” than in students who answered “completely sufficient” (crude OR = 2.03, 95% CI = 1.87–2.22). Model II was adjusted for general characteristics, and an adjusted odds ratio (AOR) was calculated, indicating that students who answered that sleep was “not at all sufficient” were at higher risk of bad breath than those who answered “completely sufficient” (AOR = 2.09, 95% CI = 1.91–2.30). Model III was adjusted for all covariates, and students who answered that sleep was “not at all sufficient” were at higher risk of bad breath than those who answered “completely sufficient” (AOR = 1.47, 95% CI = 1.33–1.83). Although the intergroup differences in Model III were observed to be lower than in Model I and Model II, this risk factor was still found to have a negative impact on bad breath in adolescents (Table 3).

## 4. Discussion

Lack of sleep is a global phenomenon in modern society, and it is a serious health problem that can have detrimental effects on children as well as adults [21].

The aim of this study was to determine whether there was a difference in the risk of bad breath according to the degree of fatigue caused by insufficient sleep reported subjectively by adolescents. The main result of this study was that adolescents who responded that their degree of fatigue recovery after sleep was “not at all sufficient” were AOR 2.09 times at higher risk of bad breath than those who responded “completely sufficient”(Model II, adjusted for general characteristics), and those who responded that the degree of fatigue recovery after sleep was “not at all sufficient” were AOR 1.47 times at higher a risk of bad breath than those who responded ‘‘completely sufficient”(Model III, adjusted for all covariates), suggesting that insufficient sleep may be a risk factor for adolescent bad breath.

Previous studies have reported that sleep deprivation is a contributing factor for various chronic diseases (diabetes, hypertension, heart disease, etc.) and is significantly related to mortality and morbidity [22]. Lack of sleep can be a contributing factor to oral disease, and long-term sleep deficiency affects two pathogenic mechanisms that can affect periodontal damage: systemic inflammation and oxidative stress; moreover, lack of sleep increases the risk of bacterial infections, which can lead to periodontal disease [22,23,24]. Saliva plays an important role in maintaining oral health, including self-cleaning, antibacterial action in the mouth, and moistening the oral mucosa. Lack of sleep during the night can reduce the amount of saliva flow in the oral cavity, which can dry the mouth, thereby increasing the number of bacteria and forming a bacterial film that can cause dental caries and bad breath [10,11,25,26]. In previous study by Suzeki et al., decreased resting salivary flow was a strong explanatory factor in clinical findings of oral malodor [25]. In another studies, chronic sleep deficiency caused daytime drowsiness, and subjects who experienced daytime sleepiness reported a decrease in saliva compared with those who did not [27,28].

Fatigue caused by lack of sleep in adolescents increases stress, and this mental fatigue eventually leads to poor oral health behaviors such as irregular brushing, leading to accumulation of dental plague and tongue coating, which are direct contributing factors to bad breath [2,4,26]. In a study by Ueno et al. (2018), the prevalence of bad breath in 520 Japanese school children was as high as 44.9%, and the risk of bad breath increased as the school year went up [29]. This rate is similar to the prevalence of bad breath in adults reported so far [29,30]. Bad breath is a major contributing factor to oral diseases such as dental caries and periodontal disease, and it can be a precursor to various systemic diseases and a sign of disease manifestation [29]. Therefore, it should not be perceived simply as an unpleasant experience for the person and others. Furthermore, bad breath should not be regarded as a mild symptom that disappears within a few days with improvements in an individual’s behavior but rather active treatment should be considered. The results of these previous studies support the results of this study whereby the risk of bad breath was increased in adolescents with insufficient sleep.

This study was a cross-sectional study and therefore could not establish the direct causal relationship between insufficient sleep and bad breath. In addition, the main variables in this study—insufficient sleep (the degree of fatigue after sleep) and bad breath—were evaluated by self-report questionnaires, which can reduce the reliability of research measurements. We were also unable to adjust compounding factors such as oral diseases (periodontal disease, dry mouth, orthodontic treatment, etc.) and diet habit (consumption of a particular food, etc.). Therefore, we could not control for all confounding variables, which may have influenced the magnitude of the study results. Therefore, more elaborate research is necessary in follow-up studies, utilizing an instrument with higher internal consistency and reliability. In addition, it is necessary to investigate the prevalence of bad breath through precise diagnosis, such as oral examination, saliva secretion test, and instrumental tests, as well as through self-reporting questionnaires to increase the reliability and validity of research tools. In addition, there have been very few studies showing that lack of sleep is a risk factor for bad breath, and there is a lack of scientific evidence to establish a direct causal relationship between the two factors. Therefore, a longitudinal study is needed to identify the causal relationship between these two factors. In the future, studies using pathway analysis or mediator analysis should be conducted that can directly or indirectly prove that insufficient sleep is the cause of oral disease, and oral disease may eventually cause bad breath.

Despite these limitations, this study has found that insufficient sleep can have a detrimental effect on bad breath in adolescents, based on a representative large-scale sample population of Korean adolescents. Therefore, the study results can be generalized to all adolescents. Further, compared with previous studies, few studies have revealed the relationship between insufficient sleep and bad breath in adolescents. Therefore, this study is valuable in terms of its novelty and scarcity. Based on the results of this study, it is necessary to recognize that insufficient sleep habits can be a risk factor for bad breath and promote proper sleeping habits in adolescents so that oral health of adolescents can be maintained through healthy sleep behaviors.

## 5. Conclusions

This study identified the relationship between insufficient sleep and bad breath in adolescents and found that adolescents who reported insufficient sleep increased their risk of bad breath. Therefore, based on the results of this study, it is necessary to recognize that insufficient sleep may be the cause of bad breath so that adolescents and their guardians can promote proper sleeping habits. In addition, it is necessary to conduct longitudinal studies to clarify the direct causal relationship and temporal sequential relationship between these two factors.

## Figures and Tables

**Table 1 ijerph-17-07230-t001:** Bad breath according to general characteristics (N = 62,276).

Variable	Category	Bad Breath N (Weighted %)		SE^a^	*p*-Value
		No	Yes		
Gender	Boys	24,486 (77.5)	7138 (22.5)	0.3	0.176
	Girls	23,955 (78.0)	6697 (22.0)	0.3	
Grade	Middle school 1st	8005 (78.3)	2184 (21.7)	0.5	0.008
	Middle school 2nd	7985 (76.7)	2392 (23.3)	0.4	
	Middle school 3rd	7976 (77.1)	2343 (22.9)	0.5	
	High school 1st	7908 (77.8)	2257 (22.2)	0.5	
	High school 2nd	8367 (77.5)	2433 (22.5)	0.4	
	High school 3rd	8200 (78.8)	2226 (21.2)	0.4	
Academic achievement	High	19,411 (79.1)	5113 (20.9)	0.3	<0.001
	Middle	14,167 (79.4)	3643 (20.6)	0.3	
	Low	14,863 (74.5)	5079 (25.5)	0.3	
Perceived family economic status	High	20,096 (80.9)	4706 (19.1)	0.3	<0.001
	Middle	22,290 (77.9)	6292 (22.1)	0.3	
	Low	6055 (67.8)	2837 (32.2)	0.5	
Living status	Living with family	46,085 (77.8)	13,077 (22.2)	0.2	0.006
	Living with relatives	377 (76.6)	119 (23.4)	1.8	
	Boarding, living in dorm (including care facility)	1979 (75.2)	639 (24.8)	0.9	
Father’s education	≤Middle school	810 (68.7)	376 (31.3)	1.4	<0.001
	High school	12,281 (77.0)	3627 (23.0)	0.4	
	University (including college)	24,801 (78.6)	6734 (21.4)	0.3	
	Unknown	8572 (77.9)	2426 (22.1)	0.4	
Mother’s education	≤Middle school	684 (68.8)	325 (31.2)	1.4	<0.001
	High school	15,026 (77.2)	4401 (22.8)	0.3	
	University (including college)	22,741 (78.3)	6284 (21.7)	0.3	
	Unknown	8260 (78.4)	2256 (21.6)	0.4	

The data were analyzed by Rao–Scott chi-square test for complex sample, SE^a^; standard error, significance level, *p* < 0.05.

**Table 2 ijerph-17-07230-t002:** Bad breath according to health behaviors (N = 62,276).

Variable	Category	Bad Breath N (Weighted %)		SE^a^	*p*-Value
		No	Yes		
**Alcohol**	No	29,882 (78.8)	7977 (21.2)	0.2	<0.001
	Yes	18,559 (76.0)	5858 (24.0)	0.3	
Smoking	No	42,311 (78.1)	11,815 (21.9)	0.2	<0.001
	Yes	6130 (75.5)	2020 (24.5)	0.5	
Stress level	Not at all	2238 (88.4)	279 (11.6)	0.7	<0.001
	Not much	8686 (84.8)	1543 (15.2)	0.4	
	A little	20,836 (79.3)	5435 (20.7)	0.3	
	A lot	12,130 (72.3)	4621 (27.7)	0.4	
	Very much	4551 (69.9)	1957 (30.1)	0.1	
Frequency tooth-brushing (per day)	None	381 (54.6))	309 (45.4)	1.9	<0.001
	Once a day	2607 (63.2)	1514 (36.8)	0.8	
	Twice a day	20,009 (75.4)	8527 (24.6)	0.3	
	3 times a day	19,250 (82.3)	4107 (17.7)	0.3	
	4 or more times	6194 (81.7)	1378 (18.3)	0.4	
Tooth-brushing after lunch	Always	12,430 (82.8)	2563 (17.2)	0.3	<0.001
	Almost all	7511 (78.5)	2044 (21.5)	0.5	
	Sometimes	10,426 (77.3)	3080 (22.7)	0.4	
	Never	18,074 (74.7)	6148 (25.3)	0.3	
Insufficient sleep	Completely sufficient	4283 (83.8)	840 (16.2)	0.5	<0.001
	Sufficient	8925 (80.9)	2100 (19.1)	0.4	
	Moderate	15,805 (78.8)	4247 (21.2)	0.3	
	Not sufficient	13,183 (75.8)	4243 (24.2)	0.4	
	Not at all sufficient	6245 (71.7)	2405 (28.3)	0.5	

The data were analyzed by Rao–Scott chi-square test for complex sample. SE^a^; standard error.

**Table 3 ijerph-17-07230-t003:** Logistic regression analysis for association between insufficient sleep and bad breath.

Variable	Model I ^a^ OR (95% CI)	Model II ^b^ AOR (95% CI) ^d^	Model III ^c^ AOR (95% CI)
Insufficient sleep			
Completely sufficient	1	1	1
Sufficient	1.22 (1.12–1.33)	1.22 (1.12–1.34)	1.10 (1.02–1.21)
Moderate	1.38 (1.28–1.50)	1.39 (1.27–1.51)	1.14 (1.04–1.24)
Not sufficient	1.64 (1.51–1.79)	1.69 (1.55–1.85)	1.30 (1.18–1.43)
Not at all sufficient	2.03 (1.87–2.22)	2.09 (1.91–2.30)	1.47 (1.33–1.83)

The data were analyzed by logistic regression for complex sample. Model I ^a^: Unadjusted odds ratio (OR) (95% CI). Model II ^b^: Adjusted for gender, grade, academic achievement, perceived family economic status, living status, father’s education, and mother’s education. Model III ^c^: Adjusted for all covariates (gender, grade, academic achievement, perceived family economic status, living status, father’s education, mother’s education, alcohol, smoking, stress level, frequency of tooth-brushing, and tooth-brushing after lunch). AOR (95% CI) ^d^: adjusted odds ratio (95% confidence Interval). Dependent variable (bad breath) reference is “No”.

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
