# Peer review of "Relationship between Insufficient Sleep and Bad Breath in Korean Adolescent Population"

_ijerph, 2020, doi:10.3390/ijerph17197230_

Round 1

Reviewer 1 Report

Thank you for the opportunity to review the manuscript entitled: 'Relationship between insufficient sleep and halitosis in Korean adolescent population' submitted to the International Journal of Environmental Research and Public Health. The cross-sectional study analysed data from the 13th Korea Youth Risk Behavior Web-based Survey to investigate the association between insufficient sleep and halitosis, among Korean adolescents. Data was obtained from 62,276, 13-18 year old respondents. Upon adjusting for a number of covariates, results suggest that those with insufficient sleep experienced higher rates of halitosis. While this study has a number of strengths, including a large sample size, and adjustment for a large number of variables, and reports some interesting findings, there are some issues that warrant further attention prior to consideration for publication. 

In the introduction, you have described the biological variables that may explain  the relationship between sleep deprivation and halitosis. It would also be of value to note some of the behavioural factors that may also explain this relationship (e.g., children who have poor sleep hygiene, may also have poor oral hygiene). 

I suggest using the term 'ethnicity' as opposed to 'race'.

I suggest making a clear distinction between halitosis and single instances of bad breath in the introduction. 

I suggest deleting the final sentence on page 1 (lines 42-43). This seems unnecessary. 

It appears that the measure of 'halitosis' may in fact, at times, be capturing isolated occurrances of bad breath.   

You note research methods as a factor that can impact on the prevalence rates of halitosis. I suggest that you provide further explanation of this point as the assumption is that it has implications for this research. 

I suggest deleting the sentence beginning 'Although halitosis should be recognised as a serious disease...' (page 2). As it is written, it disconnects the preceding and following sentences. 

Rather than referring to the National Sleep Foundation Sleep in America, I suggest simply referencing as the National Sleep Foundation. 

I suggest reporting the prevalence of halitosis in adolescents in your introduction. This will strengthen the rationale for your study by highlighting the extent of this issue. 

Please provide an overview of past research that has investigated the relationship between insufficient sleep and halitosis. This needs to be strengthened in the introduction, with supporting citations included. 

I suggest moving the paragraph beginning 'The average sleep time of adolescents...' to the previous section on sleep, and integrating it there. 

I have concerns regarding the measure of halitosis. The framing of this dependent variable would capture single, discrete episodes of bad breath, that may be the result of a number of factors (e.g., consumption of a particular food). The result of this is that is may significantly over-estimate the rates of halitosis. 

You have only used a single, self-reported, subjective measure of sleep problems. Typically, sleep most accurately assessed using a combination of objective (e.g., actigraphy) and subjective measures. 

Within your discussion (page 6, lines 217-220), I suggest moving this content to the previous section linking decreased salivary flow with bad breath. 

You focus extensively on the secondary impact of halitosis within your discussion (page 7). I suggest that you focus on this in your introduction, and too a far lesser extent in your discussion. Instead, I suggest focusing further on the implications of your findings that demonstrate a relationship between sleep and halitosis. 

On line 241, sentence including 'were evaluated by self-subjective' should read 'a single, self-report (or subjective) measure of ....'. 

Thank you again for the opportunity to review this manuscript. I wish you all the very best with your future research. 

Author Response

Reviewer 1.

Comments and Suggestions for Author

Thank you for the opportunity to review the manuscript entitled: 'Relationship between insufficient sleep and halitosis in Korean adolescent population' submitted to the International Journal of Environmental Research and Public Health. The cross-sectional study analysed data from the 13th Korea Youth Risk Behavior Web-based Survey to investigate the association between insufficient sleep and halitosis, among Korean adolescents. Data was obtained from 62,276, 13-18 year old respondents.

Upon adjusting for a number of covariates, results suggest that those with insufficient sleep experienced higher rates of halitosis. While this study has a number of strengths, including a large sample size, and adjustment for a large number of variables, and reports some interesting findings, there are some issues that warrant further attention prior to consideration for publication. 

In the introduction, you have described the biological variables that may explain  the relationship between sleep deprivation and halitosis. It would also be of value to note some of the behavioural factors that may also explain this relationship (e.g., children who have poor sleep hygiene, may also have poor oral hygiene)

I suggest using the term 'ethnicity' as opposed to 'race'

Response: Thank you for your comments. I accept your advice and revised “race” to “ethnicity”

See the line 54

 I suggest making a clear distinction between halitosis and single instances of bad breath in the introduction

Response: Thanks for the good comments. I revised to “Bad breath” consistently

I suggest deleting the final sentence on page 1 (lines 42-43). This seems unnecessary

Response: Thank you for highlighting this problem. I deleted this sentence(lines 42-43)

It appears that the measure of 'halitosis' may in fact, at times, be capturing isolated occurrances of bad breath. 

Response:  Thanks for the good comments. I think it is closer to the meaning of bad breath. Therefore, the word was modified to be bad breath consistent with the subject and content of this manuscript.

You note research methods as a factor that can impact on the prevalence rates of halitosis. I suggest that you provide further explanation of this point as the assumption is that it has implications for this research. 

Response: Various factors influencing bad breath were explained, and introduction explained that existing papers mainly deal with such factors, and the important point is that lack of sleep(fatigue after sleep) among the various factors is a factor that has rarely been dealt with in previous studies. And to prove that lack of sleep can affect bad breath. Therefore, it is believed that adding any explanation for other factors will depart from the main focus of interest in this manuscript and further lose its direction. I think my opinion has been fully explained in this manuscript. And according to the reviewer's opinion, some unnecessary sentence was deleted, and the location of the context in the flow of sentence was changed.
The manuscript was revised more concisely and convincingly by moving the contents of the discussion to the introduction and deleting unnecessary contents.

The sentence below from the discussion was moved to the introduction

The average sleep time of adolescents reported by the Korea Centers for Disease Control and Prevention (KCDCP) was much less than the sleep time recommended by The National Sleep Foundation (8.5–9.25 hours for persons aged 12–17 years, not including naps). It has been reported that sleep time decreases from elementary school to higher grades. In particular, Korean high school students report that their average sleep time is only 6.1 hours, and this is of concern and requires attention and intervention

See the lines 44-49

This sentence was deleted

“Halitosis (oral malodor) is an unpleasant odor coming from the mouth which can be unpleasant for others as well. Family members, friends, and colleagues may indirectly inform someone with bad breath by frowning or turning their faces during conversations”

I suggest deleting the sentence beginning 'Although halitosis should be recognised as a serious disease...' (page 2). As it is written, it disconnects the preceding and following sentences

Response: Thank you for your advice. We have deleted the following sentence according to your opinion

“Although halitosis should be recognized as a serious disease that should never be overlooked, it is considered a common personal symptom and is treated lightly in terms of prevention and treatment compared with other diseases”

Rather than referring to the National Sleep Foundation Sleep in America, I suggest simply referencing as the National Sleep Foundation

Response: I agree with you. I have revised as the National Sleep Foundation

See the lines 45-46

I suggest reporting the prevalence of halitosis in adolescents in your introduction. This will strengthen the rationale for your study by highlighting the extent of this issue. 

Response: I am deeply grateful for your advice. The prevalence of halitosis in the world and the prevalence of halitosis in Korean adolescents are described in the introduction part and it is also described in the discussion part

“The worldwide rate of halitosis is between 10% and 50%, and the prevalence is high in adults as well as in children and adolescent. The KYRBS (2017) reported that over 20% of adolescents answered that they had an unpleasant smell in their mouth, suggesting that the prevalence of bad breath in Korean adolescents is extremely high”.

In a study by Ueno et al. (2018), the prevalence of bad breath in 520 Japanese school children was as high as 44.9%, and the risk of bad breath increased as the school year went up”

Please provide an overview of past research that has investigated the relationship between insufficient sleep and halitosis. This needs to be strengthened in the introduction, with supporting citations included

Response: I am deeply grateful for your advice . Many literature searches have been conducted to find studies that insufficient sleep affects bad breath, but few studies have dealt with the relationship between these two factors in previous studies. In particular, no research has yet been found to prove a direct causal relationship between these two factors. This was the hardest part for me when I was writing this manuscript. So I explained that lack of sleep is a factor that can cause oral disease, that is, periodontal disease or dental caries, and indirectly explaining that it can eventually cause bad breath. In particular, it was explained that insufficient sleep can be a contributing factor in causing bad breath by reducing the flow of saliva, increasing the risk of dry mouth and infection, and accumulating plaque more actively.
In this manuscript, it has been proved that lack of sleep can affect halitosis, but the lack of scientific evidence of existing studies is a limitation, and cross-sectional studies are limited to prove a direct causal relationship. In the future, lack of sleep contributes to oral diseases, and Oral disease can eventually be a contributing factor to halitosis. Research using pathway analysis or mediating factor analysis that can be directly or indirectly proved needs to be conducted and be described as further limitations in the discussion section

See the lines 240-246, on 7page

I suggest moving the paragraph beginning 'The average sleep time of adolescents...' to the previous section on sleep, and integrating it there

Response; The sentence below from the discussion was moved to the introduction

‘The average sleep time of adolescents reported by the Korea Centers for Disease Control and Prevention (KCDCP) was much less than the sleep time recommended by The National Sleep Foundation (8.5–9.25 hours for persons aged 12–17 years, not including naps). It has been reported that sleep time decreases from elementary school to higher grades. In particular, Korean high school students report that their average sleep time is only 6.1 hours, and this is of concern and requires attention and intervention”.

I have concerns regarding the measure of halitosis. The framing of this dependent variable would capture single, discrete episodes of bad breath, that may be the result of a number of factors (e.g., consumption of a particular food). The result of this is that is may significantly over-estimate the rates of halitosis. You have only used a single, self-reported, subjective measure of sleep problems. Typically, sleep most accurately assessed using a combination of objective (e.g., actigraphy) and subjective measures. 

Response; thank you for your advice. This part is described as a limitation of the study. In addition, I described the limitations of the study in addition to this comment See the lines 232-235

Within your discussion (page 6, lines 217-220), I suggest moving this content to the previous section linking decreased salivary flow with bad breath

Response: I moved this content to the previous section linking decreased salivary flow with bad breath

See the lines 212-213

You focus extensively on the secondary impact of halitosis within your discussion (page 7). I suggest that you focus on this in your introduction, and too a far lesser extent in your discussion. Instead, I suggest focusing further on the implications of your findings that demonstrate a relationship between sleep and halitosis

Response: Thanks for your good comments.
I reviewed the contents of the discussion more carefully and deleted unnecessary contents. In addition, some of the contents were moved to the introduction, and the contents of the introduction were reinforced, and I have summarized the contents that must be covered in the discussion.

Sentence deleted from discussion:

“Bad breath can have an adverse effect on schoolmates in school life for adolescents, and the pain and psychological atrophy experienced can also have a detrimental effect on the adolescent’s mental health. In addition. (lines 226-228 on 7page, original ver.)”

Sentence moved to the introduction part:

“In a study of children and adolescents aged 12–16 years by Guedes et al. (2019), subjects with dental caries had a 3.8-fold higher risk of bad breath than those without, and decreased salivary flow increased the risk of bad breath 4.2-fold times”. (lines 217-129à lines 40-43)

On line 241, sentence including 'were evaluated by self-subjective' should read 'a single, self-report (or subjective) measure of ....'. 

Response: I revised the word “ self-subjective” to “ self-report”

Acknowledgements

Thank you for your careful review of my manuscript.
I also appreciate your advice on helping me improve my manuscript.
We will present a better paper to your journal in the future.

Reviewer 2 Report

General comments:

The topic of the paper is of interest, because both conditions are frequent in adolescents and have a negative impact on quality of life.

The data basis is huge. However, the most important confounder of an association between sleep problems and halitosis – orthodontic appliances – was not controlled for in the analyses. I expect a considerable number of subjects with orthodontic treatment in the sample. And we know, that orthodontic treatment may cause sleep problems and halitosis by plaque accumulation. This can’t be ignored.

https://doi.org/10.13065/jksdh.2014.14.03.343

https://doi.org/10.5664/jcsm.7276

Specific comments:

Table 2: The sum in the yes-category (alcohol drinking) is not 100%. However - for many reasons - it is better to report column-% instead of row-%.

Author Response

Reviewer 2. Comments and Suggestions for Authors

General comments:

The topic of the paper is of interest, because both conditions are frequent in adolescents and have a negative impact on quality of life.

The data basis is huge. However, the most important confounder of an association between sleep problems and halitosis – orthodontic appliances – was not controlled for in the analyses. I expect a considerable number of subjects with orthodontic treatment in the sample. And we know, that orthodontic treatment may cause sleep problems and halitosis by plaque accumulation. This can’t be ignored.

https://doi.org/10.13065/jksdh.2014.14.03.343

https://doi.org/10.5664/jcsm.7276

Response: Thanks for the good comments.
I also believe that orthodontic treatment may affect bad breath and sleep problems. Therefore, we tried to perform statistical analysis again by using orthodontic treatment as a covariate, but the analysis was not possible because it was not included in the questionnaire of the online  Korea Youth Risk Behavior Web-based Survey (2017). Plaque accumulation is easy if the orthodontic device is attached, but during orthodontic treatment, you should check it regularly once a month, and they will also receive oral prophylaxis and oral hygiene management education. Therefore, it is possible to overestimate the prevalence of bad breath, but on the contrary, it is thought that students who are undergoing orthodontic treatment may have better oral hygiene management than students who do not have orthodontic treatment, which may lower the prevalence of bad breath.
Therefore, as a limitation of the study, it was not possible to include orthodontic treatment as a compounding factor and it was described that this could affect the size of the study results.

Specific comments:

Table 2: The sum in the yes-category (alcohol drinking) is not 100%. However - for many reasons - it is better to report column-% instead of row-%.

Response: The gap between the variables in the table is narrow, so it is difficult to interpret the table. It seems that there were difficulties. and one value had a number notation error. I have modified that part
 colum% = 100% is correct.

I am interpreting by adjusting the spacing of the table in Table 2
It has been modified so that there is no difficulty

Thank you for your careful review of my manuscript.
I am deeply grateful to all the reviewers who reviewed my manuscript.
I believe that through your comments, my manuscript has improved considerably and the quality of the paper has improved.
Thank you.

Reviewer 3 Report

This is a cross-sectional study based on a big survey in South Korean.

Introduction and Methods are correct.

Result section: the students' parents consent the participation?

Data reported in tables is not well understood, I think is problem of the alignment of rows between columns, its difficult to see where begin the results of each main variable.

Discussion: Maybe some words about the differences of the relationship between grade and academic achievement with halitosis, also about parents' education. Limitations are well explained.

Conclusions are supported by the results.

Author Response

Reviewer 3. Comments and Suggestions for Authors

This is a cross-sectional study based on a big survey in South Korean.

Introduction and Methods are correct.

Result section: the students' parents consent the participation?

Response: The KYRBS, a government-approved survey (approval number 117058), was reviewed by the institutional review board of the Korea Centers for Disease Control and Prevention and conducted upon obtaining consent from the participants (included student’s parents).

Data reported in tables is not well understood, I think is problem of the alignment of rows between columns, its difficult to see where begin the results of each main variable.

Response: The gap between the variables in the table is narrow, so it is difficult to interpret the table. By widening the spacing of the table, it is easy to distinguish between variables. it has been modified so that there is no difficulty in understanding the table.

Thank you for your careful review of my manuscript.
I am deeply grateful to all the reviewers who reviewed my manuscript.
I believe that through your comments, my manuscript has improved considerably and the quality of the paper has improved.
Thank you.

Round 2

Reviewer 2 Report

The author presents a revised version with some improvements. The analyses do not include important confounders of the association between sleep problems and bad breath (like orthodontic appliances, ENT problems) but a lot of probably highly correlated variables (academic achievement, perceived family, economic status, living status, father's education, mother's education, alcohol, smoking, stress level, frequency tooth-brushing, tooth-brushing after lunch). This is a limitation and the author dicussed that. I think there is one (or more) factor(s) that cause both insufficient sleep and bad breath and the association between both.